# Immunotherapy for Hepatocellular Carcinoma: A 2021 Update

**DOI:** 10.3390/cancers12102859

**Published:** 2020-10-04

**Authors:** Christo Kole, Nikolaos Charalampakis, Sergios Tsakatikas, Michail Vailas, Dimitrios Moris, Efthymios Gkotsis, Stylianos Kykalos, Michalis V. Karamouzis, Dimitrios Schizas

**Affiliations:** 1First Department of Surgery, National and Kapodistrian University of Athens, Laikon General Hospital, 115 27 Athens, Greece; christo.kole@gmail.com (C.K.); mike_vailas@yahoo.com (M.V.); thymiosgotsis@yahoo.gr (E.G.); schizasad@gmail.com (D.S.); 2Department of Medical Oncology, Metaxa Cancer Hospital, 185 37 Athens, Greece; nick301178@yahoo.com (N.C.); tsakatikas.s@gmail.com (S.T.); 3Department of Surgery, Duke University School of Medicine, Durham, NC 27707, USA; dimitrios.moris@duke.edu; 4Second Propedeutic Department of Surgery, National and Kapodistrian University of Athens, Laikon General Hospital, 115 27 Athens, Greece; kykalos@gmail.com; 5Molecular Oncology Unit, Department of Biological Chemistry, National and Kapodistrian University of Athens, 115 27 Athens, Greece

**Keywords:** hepatocellular carcinoma, immunotherapy, immune checkpoint inhibitors, cancer vaccines, adoptive cellular immunotherapy, oncolytic viruses, tumor microenvironment, microsatellite instability

## Abstract

**Simple Summary:**

Hepatocellular carcinoma (HCC) is the most common liver malignancy associated with poor prognosis. Treatment options are limited partially due to resistance to traditional chemotherapeutic agents. Immunotherapy has changed the treatment landscape in metastatic and recurrent solid tumors such as malignant melanoma and non-small-cell lung cancer. Application of immunotherapy regimens in patients with HCC has led to encouraging results in terms of both safety and efficacy. In this review, we summarize the key points of currently available clinical trials and immunotherapy perspectives for HCC. Moreover, we explore the role of tumor microenvironment as a predictive and prognostic marker to immunotherapy response and its clinical implications.

**Abstract:**

Hepatocellular carcinoma (HCC) is one of one of the most frequent liver cancers and the fourth leading cause of cancer-related mortality worldwide. Current treatment options such as surgery, neoadjuvant chemoradiotherapy, liver transplantation, and radiofrequency ablation will benefit only a very small percentage of patients. Immunotherapy is a novel treatment approach representing an effective and promising option against several types of cancer. The aim of our study is to present the currently ongoing clinical trials and to evaluate the efficacy of immunotherapy in HCC. In this paper, we demonstrate that combination of different immunotherapies or immunotherapy with other modalities results in better overall survival (OS) and progression-free survival (PFS) compared to single immunotherapy agent. Another objective of this paper is to demonstrate and highlight the importance of tumor microenvironment as a predictive and prognostic marker and its clinical implications in immunotherapy response.

## 1. Introduction

Hepatocellular carcinoma (HCC) accounts for more than 80% of primary liver cancers [1]. HCC is one of the most frequent cancers and the fourth leading cause of cancer-related mortality worldwide [2] accounting for more than 800,000 deaths globally [3]. Moreover, in the next 10 years, more than one million patients are expected to die from liver cancer as estimated by the World Health Organization (WHO) [4]. The majority of HCC patients are males in a ratio of 2.4 to 1 compared to women [5]. Hepatitis B or C (HBV or HCV) is the most common risk factor, accounting for 80% of HCC cases globally [6]. Other factors that increase the risk of HCC are cirrhosis, alcohol abuse, environmental toxins, non-alcoholic fatty liver disease (NAFLD), metabolic diseases such as diabetes mellitus and obesity, smoking, and genetic and hereditary disorders [7,8]. 

The liver is considered an immune tolerant tissue, a characteristic that can be attributed to the particularities of its physiological function. Liver sinusoidal endothelial cells are exposed to a significant amount of bacterial antigens from portal circulation. These cells act as antigen-presenting cells (APCs) and regulate immunogenicity of liver microenvironment. Their role in normal liver function is to prevent acute response to bacterial agents in order to avoid unnecessary tissue damage [9]. As a result, liver sinusoidal endothelial cells express immunosuppressive molecules, such as programmed cell death ligand-1 (PD-L1). Another important cell type, Kupffer cells, are specialized liver-located macrophages that remove bacteria and produce immunosuppressive cytokines, such as IL-10 and prostaglandins [10]. They can also activate forkhead box P3 (FoxP3) in CD4+ T-cells resulting in proliferation of CD4+ regulatory T cells (Tregs), which negatively regulate immune response [11,12]. HCCs originate predominantly from hepatocytes and non-cancerous lesions (regenerative nodules and adenomas) from hepatic progenitor cells [13]. Increased PD-1 and PD-L1 expression has been observed in HCC patients [14,15], with the expression of PD-L1 associated with tumor aggressiveness and poor prognosis [16,17]. Among the molecular signaling pathways implicated in the pathogenesis of HCC, the Wnt/β-catenin signaling pathway is one of the most frequently activated [18]. Recent studies have shown that WNT/β-catenin signaling correlates with immune escape involving defective recruitment of dendritic cells by reducing CCL5 production and consequently impaired T-cell activity [19]. In addition, β-catenin activation promotes immune escape due to resistance to anti–PD-1 therapy in HCC [15,18,19]. Furthermore, high levels of Tregs in the HCC microenvironment are associated with poor prognosis. Dendritic cells (DCs), another type of APCs, also favor immune tolerance through the same mechanism [20]. Myeloid-derived suppressor cells (MDSCs) play a pivotal role in the development of immune tolerance through the expression of various cytokines (Figure 1), and their increased concentration in HCC tissues is linked to poorer prognosis [21,22]. For example, MDSCs promote tumor angiogenesis through vascular endothelial growth factor (VEGF) production and exert their immunosuppressive function through the induction of the CD4(+)CD25(+)Foxp3(+) Tregs [23,24]. MDSCs suppress natural killer (NK) cells, which limits the release of IFN-*γ* [25]. Moreover, immunosuppressive tumor-associated macrophages (TAMs) and tumor-associated neutrophils (TANs) impair CD8+ cytotoxic T lymphocytes (CTLs) [26,27], while promoting accumulation and activation of CD4+CD25+Tregs [28]. Additionally, these cells express galectin-9, a ligand to TIM-3, an immune checkpoint protein expressed on Th1 cells surface. TIM3/galectin-9 pairing has a variety of effects on T-cells function, most often leading to suppression of immune response (Figure 1) [29]. 

Diagnosis of HCC in the early-stage offers a wide array of treatment options that increase overall survival (OS) and improve quality of life. These patients can be treated with surgery [30], adjuvant or neoadjuvant chemoradiotherapy, liver transplantation, and radiofrequency ablation (RFA), albeit only one-third of patients are eligible for these approaches [31]. Unfortunately, due to late diagnosis, 70% to 80% of advanced HCC cases will not benefit from tumor resection [32]. As a result, prognosis is poor for most patients, with an average five-year survival rate of less than 15% and a median OS following diagnosis at 6 to 20 months [5,33]. Current treatment options for patients with unresectable HCC include transcatheter arterial chemoembolization (TACE) and the tyrosine kinase inhibitors sorafenib, regorafenib, and lenvatinib. These tyrosine kinase inhibitors are small molecules that inhibit multiple receptor tyrosine kinases implicated in tumor growth and angiogenesis, pathologic bone remodeling, drug resistance, and metastatic progression of cancer. HCC is a highly angiogenic tumor, thus inhibition of vascularization is a reasonable therapeutic strategy. Furthermore, inhibition of VEGF seems to enhance immunotherapy efficacy by exercising an immunomodulatory role in the tumor microenvironment. However, clinical benefit is still limited, and new therapeutic modalities are being explored [34,35,36]. 

Immunotherapy has proven to be effective and safe in treating a plethora of solid tumors, prolonging OS, and offering a tolerable toxicity profile [37,38,39]. Immunotherapy negates tumor-expressed extracellular ligands that suppress intrinsic immune response. Examples of such molecules are cytotoxic T-lymphocyte-associated antigen 4 (CTLA-4), programmed cell death protein-1 (PD-1), and its ligand, PD-L1. These proteins prevent T cells from recognizing and eliminating cancer cells [40]. This allows regular cells to avoid autoimmune destruction by downregulating T-cell activation [41]. CTLA-4 competitively inhibits binding of B7 ligands to the co-stimulatory receptor CD-28, while PD-1 binds to PD-L1 and PD-L2 ligands, preventing T-cell activity in peripheral tissues [41]. Overexpression of PD-L1 has been detected in the microenvironment of several solid tumors, such as esophageal, colon, pancreatic, gastric, lung, breast [42], and HCC [43]. Checkpoint inhibitors are antibodies that activate T-cell mediated antitumor responses by selectively blocking the checkpoint receptors PD-1, PD-L1, and CTLA-4 [44]. Targeting one or more of these receptors could mediate tumor regression in patients with melanoma, lung cancer, renal cell carcinoma, urothelial cancer, head and neck cancer, and other malignancies [45].

On the other hand, therapeutic cancer vaccines use a tumor-associated antigen (TAA) originating either from whole-cell tumor lysates and recombinant tumor peptides or full-length proteins or recombinant viruses encoding for TAAs. TAAs are transferred and presented by major histocompatibility complex (MHC) class I molecules in APCs to effectively induce activation of cytotoxic T-lymphocytes (CTLs) [46,47]. Another strategy in immune-regulated antitumor response is that of adoptive cell transfer (ACT). Immune cells are extracted from patients’ peripheral blood and undergo genetic engineering to express chimeric antigen receptors (CARs). These cell membrane proteins bind to specific cancer antigens stimulating immune destruction of tumor cells [48].

## 2. Immune Checkpoint Inhibitors in Hepatocellular Carcinoma 

Immune checkpoint inhibitors (ICIs) are monoclonal antibodies that block extracellular proteins that suppress antitumor immune response. Both tumor and immune system cells express these ligands. Although many molecules have been identified as mediating immune evasion by cancer cells, two categories have been thoroughly examined in clinical trials, PD-1 and CTLA-4 [49]. Currently, the US Food and Drug Administration (FDA) has approved checkpoint inhibitors for use in HCC (Table 1) [50], but many more promising markers are being investigated in animal models, and new agents are being tested in clinical trials. Lymphocyte activation gene 3 (LAG-3) is a membrane protein closely related to CD4. It is expressed by a variety of T cells, such as CD4, CD8, and Tregs, as well as by NK cells, DCs, and B cells. LAG-3 binds to MHC II of APCs and prevents recognition by T-cell receptors (TCRs), thus suppressing T-cell mediated immune response. LAG-3 expression is usually accompanied by increased PD-L1 levels in tumor tissue [51]. As a result, development of LAG-3 inhibitors and combination with existing anti-PD-1/PD-L1 molecules could have significant synergistic clinical benefit. However, no clinical trials are currently ongoing on HCC using these targets.

### 2.1. Nivolumab

Nivolumab is a human anti-PD-1 IgG4 monoclonal antibody that blocks PD-1 and was approved by the FDA in September 2017 as second-line treatment for HCC after progression of disease on first-line therapy with sorafenib. Several ongoing clinical trials are exploring the effectiveness and safety of nivolumab in patients with HCC [52]. In an open-label, non-comparative, phase I/II dose-escalation and expansion trial CheckMate 040 (NCT01658878), patients with a Child–Pugh score of 7 or less received 0.1 mg/kg to 10 mg/kg of nivolumab every 2 weeks (dose-escalation phase) and those with a Child–Pugh score of 6 or less, received 3 mg/kg every 2 weeks (dose-expansion phase). Child–Pugh is a score designed to assess the prognosis of chronic liver disease, primarily cirrhosis and to predict mortality, requirement of strength of treatment, and necessity of liver transplant [61]. Overall, this regimen resulted in substantial tumor reductions and had a manageable toxicity profile (Table 1). Patients in the dose-escalation phase achieved a median OS of 15 months (95% CI: 9.6–20.2). Both groups performed similarly in terms of objective response rate (ORR). In the dose-escalation and dose-expansion group, ORR was 15% (95% CI: 6–28) with a median duration of response at 17 months and 20% (95% CI: 15–26) with median duration at 9.9 months, respectively [52]. The durable objective responses showed the potential of nivolumab for treatment of advanced HCC. Immunohistochemistry and RNA sequencing analysis revealed that PD-1- and PD-L1-positive patients were associated with improved survival and response. Patients with tumor PD-L1 ≥1% showed an increased median OS of 28.1 months (95% CI 18.2-n.a.) compared to 16.6 months for those with tumor PD-L1 <1% (95% CI 14.2–20.2). Moreover, macrophage markers were not associated with OS, and increased CD3+ and CD8+ T-cells showed a non-significant trend towards improved OS, while patients with baseline AFP <400 μg/L demonstrated numerically improved median OS of 16.8 months (95% CI 13.3–20.2) compared with a median OS of 13.0 months (95% CI 8.0–17.5) in patients with AFP ≥400 μg/L [62].

Another randomized phase III study, CheckMate 459 (NCT02576509), evaluated the efficacy of nivolumab vs. sorafenib, a tyrosine kinase inhibitor, as a first-line treatment [53]. OS in nivolumab-treated patients did not meet the predefined threshold of statistical significance (HR 0.84, *p* = 0.0419). However, a clinically meaningful improvement in median OS of 16.4 months (95% CI: 13.9–18.4) vs. 14.7 months (95% CI: 11.9–17.2) for sorafenib-treated patients was demonstrated. ORRs in the nivolumab and sorafenib arms were 15% and 7%, respectively. Interestingly, in the nivolumab arm, patients with PD-L1 > 1% benefited the most compared to those with PD-L1 < 1%, ORR of 28.2% vs. 12.2%, respectively. However, this did not translate to improved OS and progression-free survival (PFS). Nivolumab was, in general, more tolerable, with 22% of patients in the nivolumab arm manifesting grade 3/4 treatment-related adverse events compared to 49% in the sorafenib arm, demonstrating a favorable safety profile consistent with previous reports [53]. The role of nivolumab in HCC treatment is currently being explored in clinical trials, either as monotherapy or in combination with other modalities. Notable examples are a phase II/III study (NCT04268888) in addition to TACE [63], phase I/II studies (NCT02423343, NCT03893695, NCT03059147) in addition to a novel transforming growth factor-beta (TGF-β) receptor I kinase inhibitors [64,65,66] and as adjuvant therapy (NCT03383458, NCT03572582) in patients at high risk of recurrence after resection or RFA compared to placebo [67,68]. 

### 2.2. Pembrolizumab

Pembrolizumab is an anti-PD-1 IgG4 antibody that was granted accelerated approval by FDA in November 2018 as a second-line treatment after progression or high toxicity with previous sorafenib, after showing increased effectiveness and tolerability in the KEYNOTE-224 phase II study (NCT02702414) [54]. The ORR was 17% (95% CI: 11–26), with 1% of patients showing complete and 16% partial response. Meanwhile, 44% of patients had stable disease, and 33% had progressive disease. Median OS reached 12.9 months (95% CI: 9,7–15,5) and median PFS 4.9 months (95% CI: 3,4–7,2). Disease control was reported in 64 of the 104 treated participants, 62% (95% CI: 52–71), while grade 3/4 treatment-related adverse events occurred at 25%. One death associated with ulcerative esophagitis was attributed to treatment [54]. Further assessment is ongoing in the phase III studies KEYNOTE-240 [55] and KEYNOTE-394 [69]. In the KEYNOTE-240 study (NCT02702401), 413 patients were recruited (279 patients received pembrolizumab and 134 placebo). Even though the results of this study did not meet the prespecified cut-offs for statistical significance, patients treated with pembrolizumab demonstrated a better ORR, 18.3% (95% CI: 14.0–23.4), compared to 4.4% (95% CI: 1.6–9.4) in the placebo group, per response evaluation criteria in solid tumors version 1.1 (RECIST 1.1). OS was determined at 13.9 months (95% CI: 11.6–16.0) and 10.6 months (95% CI: 8.3–13.5) for pembrolizumab and placebo groups, respectively, with a hazard ratio (HR) estimated at 0.7 (95% CI: 0.611–0.998, *p* = 0.023) (Table 1) [55]. Safety and efficacy of pembrolizumab in combination with sorafenib is being evaluated in a randomized phase I/II (NCT03211416) study [70], while KEYNOTE-394 (NCT03062358), a phase III study evaluating the effectiveness of pembrolizumab as second-line therapy after progression or intolerance to first-line sorafenib or oxaliplatin-based chemotherapy in Asian patients [69], are currently on recruitment phase and no results are published yet. 

An open-label multicenter Phase Ib study reported that combination of pembrolizumab with lenvatinib (a multiple kinase inhibitor against VEGFR1, VEGFR2, and VEGFR3 kinases) had promising antitumor activity (Table 1) and manageable toxicities with grade ≥ 3 treatment-related adverse events reported in 67% and grade 5 to 3% of patients [56]. Patients with unresectable HCC received lenvatinib (bodyweight ≥ 60 kg, 12 mg; <60 kg, 8 mg) orally daily and pembrolizumab 200 mg intravenously on day 1 of a 21-day cycle. Authors reported a median OS at 22 months with a PFS at 9.3 months per modified RECIST (mRECIST) and 8.6 months per RECIST 1.1 [56]. Safety and efficacy of pembrolizumab in combination with lenvatinib is also evaluated in a randomized, double-blind, phase III (NCT03713593) study [71]. Moreover, pembrolizumab is also currently evaluated as adjuvant therapy after RFA or radiotherapy (NCT03753659, NCT03316872, NCT03099564, NCT03939975) [72,73,74,75].

### 2.3. Atezolizumab

Atezolizumab is an engineered IgG1 mAb targeting PD-L1. A randomized, phase Ib study (NCT02715531), in unresectable HCC patients, showed a manageable toxicity profile and a significantly better PFS in patients receiving atezolizumab plus bevacizumab compared to atezolizumab as monotherapy; 5.6 months (95% CI: 3.6–7.4) vs. 3.4 months (95% CI: 1.9–5.2), respectively [HR 0.55 (80% CI: 0.40–0.74); *p* = 0.0108] (Table 1). Grade 3 or 4 adverse events occurred in 20% and 5% for atezolizumab/bevacizumab and atezolizumab as monotherapy, respectively [57]. The recent, open-label, phase III, IMbrave150 clinical trial (NCT03434379) further evaluated the effect of atezolizumab/bevacizumab in 336 patients compared to treatment with sorafenib in 165 patients [58]. This study presented a significantly lower HR for death in atezolizumab/bevacizumab treated patients compared to sorafenib [HR 0.58 (95%CI: 0.42–0.79); *p* < 0.001] (Table 1). Moreover, the combination of atezolizumab/bevacizumab significantly improved the 12-month OS rate to 67.2% (95% CI: 61.3–73.1) compared to 54.6% (95% CI: 45.2–64.0) in the sorafenib group. Median PFS was also significantly better; 6.8 months (95% CI: 5.7–8.3) and 4.3 months (95% CI: 4.0–5.6), respectively [HR 0.59 (95% CI: 0.47–0.76); *p* < 0.001]. Moreover, Finn et al. reported an improvement in quality of life as expressed by median time to disease deterioration; 11.2 months (95% CI: 6.0–NE) vs. 3.6 months (95% CI: 3.0–7.0) [HR: 0.63 (95% CI: 0.46–0.85)] and grade 3 or 4 adverse events occurring at 56.5% and 55.1% for atezolizumab/bevacizumab and sorafenib, respectively [58]. Following these encouraging results, the combination of atezolizumab/bevacizumab is a promising treatment option in previously untreated, unresectable HCC and gained FDA approval in the first line (National Comprehensive Cancer Network category 1 recommendation). Additionally, a COSMIC-312 Phase III study (NCT03755791) is designed to review the effect of atezolizumab plus cabozantinib (an oral tyrosine kinase inhibitor that targets VEGFR, MET, AXL, KIT, FLT-3, Tie-2, and RET) versus sorafenib in treatment-I advanced HCC; however, no results are yet available [76,77]. Previous analysis of cabozantinib improved the primary end point of OS relative to placebo, with a median of 10.2 versus 8.0 months (hazard ratio [HR] 0.76; 95% CI 0.63–0.92; *p* = 0.005), and the secondary end point of PFS, with a median of 5.2 versus 1.9 months (HR: 0.44; 95% CI: 0.36–0.52; *p* < 0.001) [76]. Furthermore, two phase III clinical trials are currently recruiting patients using atezolizumab plus bevacizumab in combination with TACE or as adjuvant therapy to resection/RFA (NCT04102098, NCT04224636) [78,79].

Several ongoing studies are using anti-PD-1 antibodies, such as Tislelizumab (BGB-A317) compared to sorafenib (NCT03412773), SHR-1210 (NCT04297202, NCT02989922), and anti-PD-L1 durvalumab (NCT03847428, NCT03970616, NCT03778957, NCT04124991), avelumab (NCT03475953), currently being evaluated in a phase I/II or III clinical study either as first-line or second-line monotherapy [80,81,82,83,84,85,86] or in combination with other inhibitors as well as with locoregional treatment (NCT04310709, NCT03869034, NCT03794440, NCT03764293, NCT03755739, NCT03847428, NCT04273100, NCT03857815) [68,87,88,89,90]. However, none of these studies have posted any results yet.

### 2.4. Tremelimumab

Tremelimumab is a human IgG2 anti-CTLA-4 inhibitor [91]. A pilot study (NCT01008358) using tremelimumab studied the toxicity and tumor response in HCC patients. Most patients had advanced-stage disease with Child–Pugh score class B. A partial response rate at 17.6% and disease control rate at 76.4% was observed. Sangro et al. reported that the time to progression was 6.5 months (95% CI: 3.95–9.14), with the treatment being, in general, well-tolerated [59]. In another study (NCT01853618), the safety and feasibility of tremelimumab combined with RFA or TACE were evaluated in patients with advanced HCC [60]. The primary results showed that 26.3% (95% CI: 9.1–51.2%) of patients achieved a confirmed partial response while the 6- and 12-month tumor PFS were reported at 57.1% and 33.1%, respectively. Median time to tumor progression of 7.5 months (95% CI: 5.6–9.3) and a median OS of 8.4 months (95% CI: 6.5–10.3) were reported, concluding that the combination of tremelimumab and RFA or TACE may be a potential new treatment option for HCC patients [60].

The efficacy of immunotherapy can be improved through combinations with chemotherapy and local disease control interventions, such as TACE and RFA. Lysis of tumor and tumor-suppressive cells causes the release of TAAs into the tumor microenvironment (TME), thus inducing a Th1 immune response by sensitizing local CD8+ T-cells and DCs. This synergy, a.k.a. the abscopal effect, has been explored in clinical trials, showing improved clinical outcomes of immune-modulating agents, especially vaccines and CTLA-4 inhibitors, when combined with other treatment modalities [92].

## 3. Vaccine Therapy in Hepatocellular Carcinoma

Therapeutic vaccines include peptides, DCs, whole-cell vaccines, oncolytic viruses, and DNA agents to increase or achieve specific immune responses to tumor antigens [47]. Regarding this, several peptides, such as alpha-fetoprotein (AFP), multidrug resistance-associated protein 3 (MRP3), and glypican-3 (GPC3), have been examined to date (Table 2) and have proved to be well-tolerated and safe.

### 3.1. Alpha-Fetoprotein (AFP) Peptide

Alpha-fetoprotein (AFP) peptide is a 70 KDa transporter, primarily expressed in the embryonic yolk sac of developing fetus and in the liver. Serum AFP levels become almost undetectable after birth; however, levels rise in HCC, and therefore, AFP is used as a biomarker [100]. Butterfield et al. used human AFP peptide epitopes previously identified [101] and created a human AFP-expressing replication-deficient adenovirus as a potential target for T-cell-based immunotherapy [102]. Therefore, these AFP constructs were tested as part of a phase I/II trial (NCT00093548) in two HCC patients who had an AFP-expressing tumor and previous treatment for HCC. This clinical trial showed that the vaccine was well-tolerated and safe, with no clinically significant adverse events. Moreover, both patients showed immunologic evidence of immunization with the AFP-specific CD8+ T cells appearing high. The first patient showed an AFP-specific T-cell response at 9 months while the second patient developed a strong AFP-specific CD8^+^ and CD4^+^ cellular response and an AFP-expressing replication-deficient adenovirus (AdV) neutralizing antibody response after 18 months [103].

### 3.2. Glypican-3 (GPC3)

Glypican-3 (GPC3) is a protein overexpressed in HCC tissues, but not in the healthy adult liver [104]. Various immunotherapies targeting glypican-3 have been developed so far (Table 2). In a phase I clinical study (UMIN000001395), GPC3-derived peptide vaccine was used in 33 patients with advanced HCC and reported that vaccination was well-tolerated, inducing a high rate of GPC3-specific CTL response [93]. One patient showed a partial response, and 19 patients showed stable disease 2 months after initiation of treatment. Furthermore, increased GPC3-specific CTLs, following vaccination correlated with significantly improved median OS of 9.0 months (95% CI: 8.0–10.0) compared to patients who had low numbers of GPC3-specific CTLs [93]. Another phase II study showed that GPC3-positive patients treated with adjuvant vaccination had significantly lower recurrence rates than patients who received surgery only (24% vs. 48%, *p* = 0.047) at 1 year and (52.4% vs. 61.9%, *p* = 0. 387) at 2 years [94].

### 3.3. Multidrug Resistance-Associated Protein 3 (MRP3)

Multidrug resistance-associated protein 3 (MRP3) is a carrier-type transport, member of ATP-binding cassette (ABC) transporters, and its high expression is related to various cancer cells [105]. Mizukoshi et al. reported an increase in MRP3 expression level in HCC tissue, significantly higher than in non-cancerous tissue (*p* < 0.05) [106]. MRP3-specific CTLs can be activated regardless of liver function, HCV infection status, AFP levels, and the stage of HCC. Moreover, Tomonari et al. demonstrated that MRP3 plays a vital role in resistance to sorafenib toxicity in HCC cells [107]. Thus, MRP3 consists a potential candidate for tumor antigen with strong immunogenicity in HCC immunotherapy. A phase I clinical trial (UMIN000005678) investigated the safety and immunogenicity of an MRP3-derived peptide (MRP3765) as a vaccine in 12 HLA-A24-positive HCC patients [95]. The vaccination was well-tolerated, inducing immunization in 72.7% of patients with the median OS being 14.0 months (95% CI: 9.6–18.5). OS was longer compared with that in studies, including patients treated with hepatic arterial infusion chemotherapy without peptide vaccination, median OS 12.0 to 12.6 months [95,108]. 

### 3.4. NY-ESO-1 and MAGE-A

NY-ESO-1 and MAGE-A: The New York esophageal squamous cell carcinoma-1 (NY-ESO-1) and the melanoma-antigen family A (MAGE-A) are two cancer-testis antigens that represent promising targets due to low expression in healthy tissue [109]. Flecken et al. reported specific CD8+ T-cell responses to NY-ESO-1b in 48% of patients with NY-ESO-1 mRNA-positive HLA-A2-positive HCC. Moreover, the presence of these responses correlates with patient survival [110]. On the other hand, MAGE-A expression profile of HCC reports that 92.3% of the tumors expressed one MAGE-A gene [111], while another study reported that MAGE/tetramer+ CD8 cells of patients with HCC are able to recognize the MAGE-1 sequence 161-169 and the MAGE-3 sequence 271-279 [112]. These results lead to the conclusion that MAGE-A antigens may represent useful targets for tumor-specific immunotherapy in HCC patients, in addition to established treatment options. However, no studies have examined the clinical response using either NY-ESO-1 or MAGE-A vaccines in patients with HCC. 

### 3.5. Dendritic Cell Vaccine

Dendritic cell vaccine: DCs are APCs, responsible for T-cell stimulation and antitumor immune response enhancement [113]. DCs are injected back into the patient after maturation and activation with a specific antigen in vitro. Studies using tumor cell lysate-loaded dendritic cell vaccine have shown antitumor effects in murine models [114]. At the same time, DC-derived exosomes form a new class of vaccines for cancer immunotherapy that can trigger potent antigen-specific antitumor immune responses and reshape the tumor microenvironment [115].

A phase I study of autologous dendritic cell-based immunotherapy was performed in unresectable primary HCC patients to evaluate the safety and feasibility of immunotherapy [116]. Eight out of ten patients included in this study had HCC, whereas the rest suffered from cholangiocarcinoma. The authors reported that immunization was well-tolerated in all patients, and no significant toxicity was detected. Moreover, one patient achieved tumor shrinkage and showed necrotic change on computed tomography, while in two other patients, serum levels of tumor markers decreased after vaccination. Another phase II clinical trial of 35 patients using DCs, pulsed ex vivo with a liver tumor cell line lysate (HepG2), resulted in generation of antigen-specific immune responses in some cases, while administration of these modified DCs was safe and well-tolerated with evidence of antitumor efficacy [96]. The median survival of the 35 treated patients was 5.5 months, while 6-month and 1-year survival rates were 33% and 11%, respectively [96]. A phase I trial (NCT01974661) confirmed the safety of intra-tumoral injection of ilixadencel (pro-inflammatory allogeneic DCs stimulated by GM-CSF and IL-4), either as monotherapy or in combination with sorafenib, and was associated with increased tumor-specific CD8^+^ T cells. Rizell et al. reported that the median time to progression was 5.5 months, and OS ranged from 1.6 to 21.4 months (Table 2). The Kaplan–Meier median OS times were 7.5 months overall, 7.4 months for the dose of 10 × 10^6^ viable cells, and 11.8 months for the dose of 20 × 10^6^ viable cells [97]. Moreover, in combination with TACE, DC infusion enhances the tumor-specific immune responses more effectively than TACE alone, although the effect is not sufficient to prevent HCC recurrence [117]. Further clinical trials are ongoing, NCT01821482, NCT02638857, NCT02882659, NCT03674073, NCT03203005 [118,119,120,121]; however, results have not been presented so far.

### 3.6. Oncolytic Viruses

Oncolytic viruses are viral particles engineered to cause direct lysis of tumor cells, resulting in the release of soluble cancer antigens, which induce antitumor neoantigen-specific CTL responses [122,123]. A randomized phase II clinical trial (NCT00554372) studied the feasibility of two doses of JX-594 (Pexa-Vec), an oncolytic virus, in 30 HCC patients by infusing low- or high-dose JX-594 into tumors. Heo et al. reported significantly longer median OS in the high-dose arm compared to the low-dose arm, 14.1 months and 6.7 months, respectively (HR 0.39, *p* = 0.020) (Table 2). The most common adverse reaction was flu-like syndrome with fever, rigor, and vomiting, which occurred in all patients within the first few days after treatment in a dose-dependent manner [98]. However, in patients who had been previously treated with sorafenib (NCT01387555), the median OS was not significantly different in patients treated with JX-594, 4.2 months, compared to best supportive care, 4.4 months, [HR 1.19 (95% CI: 0.78–1.80); *p* = 0.428) [99]. Currently, a phase III study (NCT02562755) is ongoing comparing JX-594, followed by sorafenib versus sorafenib alone [124].

## 4. Adoptive Cell Transfer in Hepatocellular Carcinoma

ACT, including NK cells, tumor-infiltrating lymphocytes (TILs), cytokine-induced killer cells (CIKs), and CAR T-cell therapy has shown considerable antitumor effects on HCC in several clinical trials (Table 3) [125,126,127]. Moreover, Zerbini et al. reported an increased NK-cell response in HCC patients after RFA [128]. Currently, only two phase II clinical trials evaluate safety and efficacy of autologous NK-cell reinfusion after curative liver resection (NCT01147380 [129] and NCT02008929 [130]).

### 4.1. CIK Cells

CIK cells: A phase I clinical study using autologous TILs following tumor resection in HCC patients reported no major adverse events after a median follow-up of 14 months. All patients remained alive, and 80% of them showed no evidence of disease [131]. The benefit of CIK cell treatment is supported by several other clinical studies [132,133,134,135,136], as well as a systematic review/meta-analysis including 13 phase II/III studies demonstrating significant superiority in prolonging the median OS, PFS, ORR, and disease control rate in HCC patients [137]. A phase III clinical study (NCT01749865) is completed; however, no results are yet available. Briefly, a phase II study (127 patients), a phase III study (200 patients), and a retrospective study (410 patients) reported similar results in patients who were treated with CIK cells as postoperative adjuvant therapy compared to no postoperative adjuvant therapy [132,133,134]. All studies reported significantly higher disease-free survival rates compared to control but no statistically significant difference in OS [132,133]. Pan et al. concluded that CIK-treated groups displayed significantly better OS compared to surgery alone in patients diagnosed with more than 5 cm tumors (*p* = 0.0002), while patients treated with more than eight cycles of cell transfusion showed significantly better OS than those treated with less than eight cycles (*p* = 0.0272). These results indicate that patients with large tumors might benefit more from CIK cell adjuvant treatment than patients with small tumors [133]. 

Combination of CIK immunotherapy with minimally invasive therapies for patients without previous surgery represents a potentially safe treatment modality for HCC [138,139,140]. Patients who had not previously received any surgery or chemoradiotherapy received CIK cells combination with TACE and RFA (CIK+TACE+RFA) (Table 3). No significant differences in disease control rates were reported between CIK cells’ treatment combined with TACE and/or RFA or TACE and RFA alone. However, Kaplan–Meier analysis showed that patients in the CIK+TACE+RFA group compared to TACE+RFA alone had significantly longer OS; 56 months (95% CI: 38.09–73.91) compared to 31 months (95% CI: 24.53–37.47) and PFS at 17 months (95% CI: 10.96–23.04) compared to 10 months (95% CI: 8.57–11.44) [138]. 

### 4.2. Chimeric Antigen Receptor T Cells (CAR-T)

Chimeric antigen receptor T cells (CAR-T): Recent studies of HCC tumor xenografts in mice and in vitro demonstrated that engineered CAR-T cells expressing a GPC3 CAR could eliminate GPC3-positive HCC cells [141,142,143,144]. Therefore, phase I clinical studies designed to evaluate the safety and efficacy of CAR-GPC3 T-cell therapy alone (NCT03980288, NCT04121273, NCT03884751) or in combination with cyclophosphamide and fludarabine (NCT02905188) or other treatment options (NCT04093648, NCT03198546) are currently ongoing and in recruiting status [145,146,147,148,149,150]. The NCT02905188 and NCT03146234 studies (Table 3) reported that patients with advanced GPC3+ HCC (Child-Pugh A), receiving autologous CAR-GPC3 T-cell therapy following cyclophosphamide and fludarabine had a tolerable toxicity profile with no grade 3/4 neurotoxicity. The OS rates at 6 months, 1 year, and 3 years were 50.3%, 42.0% and 10.5%, respectively, with a median OS of 9.1 months (95% CI: 1.5–20) [151]. The target lesions in two patients with partial response exhibited significant tumor shrinkage, while one patient with sustained stable disease was alive after 44.2 months [151]. Another phase I study (NCT02395250) [152] and a phase I/II study (NCT02723942) [153] have been completed; however, no results are yet posted. Moreover, clinical trials using CAR-T cells targeting other antigens are currently ongoing (NCT02587689, NCT03013712) [154,155].

A study published in Lancet in 2000, which followed 150 patients for 4.4 years, assigned 76 patients for adoptive immunotherapy and 74 patients for no adjuvant treatment and concluded that adoptive immunotherapy is a safe and feasible treatment that can lower recurrence after surgery for HCC [156]. Takayama et al. showed that adoptive immunotherapy decreased recurrence frequency by 18% compared with controls (no adjuvant treatment) and reduced the risk of recurrence by 41% (95% CI: 12–60, *p* = 0.01). The immunotherapy group had a significantly longer recurrence-free survival (*p* = 0.01) and disease-specific survival (*p* = 0.04) than the control group; however, no difference in median OS was identified between groups (*p* = 0.09) [156]. 

## 5. Combinations Strategies of Immunotherapies

Combination of nivolumab (NIVO) with ipilimumab (IPI) in sorafenib-treated patients has shown clinically significant responses and had an acceptable safety profile, with an ORR twice that of NIVO mono (31% and 14%, respectively) [157]. Patients were randomized to three groups (Table 4)—(A) NIVO (1mg/kg) + IPI (3mg/kg) (four doses), (B) NIVO (3mg/kg) + IPI (1mg/kg) (four doses), or (C) NIVO (3mg/kg) + IPI (1mg/kg). Overall, ORR was 31% with a median 24-month OS rate at 40%. Patients in arm A, B, and C had a median OS of 23 (95% CI: 9–NA), 12 (95% CI: 8–15) and 13 months (95% CI: 7–33), respectively. Combination of NIVO + IPI was well-tolerated; 37% of patients had a grade 3–4 treatment-related adverse event while 5% had grade 3–4 leading to discontinuation. Another study, reported combination therapy with nivolumab, ipilimumab, and cabozantinib (CABO) led to clinically meaningful responses (Table 4). Median PFS was 5.5 months for the patients receiving NIVO (240mg/day) + CABO (40mg/day) and 6.8 months for the NIVO (3mg/kg) + IPI (1mg/kg)+ CABO (40mg/day), while median OS was not reached in either arm. Grade 3–4 treatment-related adverse events were reported in 15 pts (42%) in the NIVO + CABO arm and 25 pts (71%) in the NIVO + IPI + CABO arm and led to discontinuation in 1 (3%) and 7 (20%) patients, respectively [158]. Several other phase I/II clinical trials evaluating safety and efficacy of nivolumab in combination with ipilimumab are currently underway (NCT03682276, NCT04039607) [159,160,161], (NCT03510871, NCT03222076) [162,163]. Moreover, checkpoint inhibitors are further combined with oncolytic viruses in currently ongoing clinical trials (NCT03071094, NCT02432963, NCT04251117, NCT04248569) [164,165,166,167]; however, no results have been reported yet. Moreover, a multi-center, global, phase III study (NCT03298451) and another two, combining durvalumab/tremelimumab with TACE or RFA (NCT02821754, NCT03482102) are currently recruiting patients [168,169]. 

Results have been reported so far from a phase I/II, open-label, randomized study (NCT02519348), concerning safety and efficacy of the recommended phase II doses of the durvalumab/tremelimumab combination for patients with HCC. In general, the treatment was well-tolerated, and no corticosteroids were required for severe immune-mediated adverse events, while enrollment to the phase II part of the study is ongoing [170]. Moreover, a multi-center, global, phase III study (NCT03298451) and another two combining durvalumab/tremelimumab with TACE or RFA (NCT02821754, NCT03482102) are currently recruiting patients [168,169]. Another study (UMIN000005820) reported that HCC patients treated with an autologous tumor lysate-pulsed DC vaccine combined with activated T cell transfer (ATVAC) resulted in improvement of median PFS and OS, 24.5 months (95%CI: 7.8–41.2) and 97.7 months (95% CI: 48.6–146.7), respectively, compared to 12.6 months (95% CI: 6.9–18.3) and 41.0 months (95% CI: 16.3–65.8) in the group receiving surgery alone (Table 4). In the treated group, patients with positive delayed-type hypersensitivity (DTH) had a better prognosis (PFS, *p* = 0.019; OS, *p* = 0.025). No adverse events of grade 3 or more were observed [171].

## 6. Hepatitis Infection and Immunotherapy

As previously mentioned, most cases of HCC are associated with chronic hepatitis infection. Thus, TME in HCC usually presents elements of chronic inflammation. Liver tissue in patients with HCC has a high concentration of CD8+ CTLs [172]; however, they are suppressed and proliferate at a reduced rate. This T-Cell exhaustion is linked to overexpression of co-stimulatory molecules, such as B7-1 (CD80) and B7-2 on CD4+ T-cells and CD137 on CD8+ and NK cells. These molecules are vital points in the signaling of immune checkpoint pathways B7-CD28/CTLA-4 and PD-L1/PD-L2/PD-1 [173]. This immunosuppressive TME prevents tumor antigen detection by DCs via suppression of TAA and MHC molecules through the expression of inhibitory molecules (IL-10, TGF-β, VEGF) (Figure 1) [174]. Other important extracellular ligands that suppress antitumor immune response are TIM-3 and LAG-3 proteins and mutated MHC type I receptors [175]. The existence of a high amount of exhausted CD8+ T-cells that overexpress negative co-stimulatory molecules, such as PD-L1, represents a clue for the use of ICIs in HCC. A high level of PD-L1 is also an adverse prognostic factor, thus increasing the need for effective treatment in this group of patients [176]. Several clinical trials have evaluated the safety and efficacy of anti-PD-L1 in HCC patients with hepatitis, though no data have been powered for statistical comparison.

In the CheckMate 040 study, ORRs with nivolumab in patients infected with HCV, HBV, and those without viral hepatitis were 20%, 14%, and 22%, respectively. In the dose-expansion phase, 6- and 9-month OS rates were 84% and 70% in HBV+ patients, 85% and 81% in HCV+, while for the entire population of the study, 83% and 74%, respectively [52]. In the CheckMate 459 study (nivolumab vs. sorafenib), a consistent effect on OS was also observed in advanced HCC with nivolumab, and benefit was noted for patients with HBV infection [53,177]. 

Pembrolizumab, in the KEYNOTE-224 study, ORR was 13% in the subgroup of HBV+ or HCV+ and 20% in the uninfected subgroup; however, this difference was not statistically significant. Reduction from baseline in tumor target lesion size was 50% in the overall study population, 58% in uninfected patients, 57% in HBV+, and 39% in HCV+ patients [54]. In the phase 3 KEYNOTE-240 study, ORR in the whole population was 18% for pembrolizumab vs. 4% for placebo, while HBV+ patients achieved improved OS in comparison to placebo [HR 0.57 (CI: 0.35–0.94)], while no significant benefit appeared in HCV+ patients [55]. In the phase III, IMbrave150 study, combination of atezolizumab and bevacizumab prolonged median PFS of HBV+ HCC compared to sorafenib, but this phenomenon did not appear in the population of HCC of non-viral etiology (median PFS, HBV+ HCC: 6.7 vs. 2.8 months; non-viral HCC: 7.1 vs. 5.6 months) [177].

A pooled analysis of previous studies by Li et al. assessed the efficacy of PD-1/PD-L1 blockade, either as monotherapy or in combination with other agents, in HCC patients according to their HBV status. No statistically significant difference was found between the two groups [odds ratio (OR) 0.68; 95% CI: 0.37–1.25], in terms of ORR, both in monotherapy and in combined therapy subgroups; however, HBV uninfected patients enjoyed statistically significant better disease control rates (DCRs) in comparison to those with HBV+ status (OR 0.49; 95% CI: 0.27–0.89 in the monotherapy subgroup vs. OR 0.52; 95% CI: 0.27–0.99 in the combination subgroup). HBV+ patients were also compared with HCV+ in the same studies, and no significant difference was in ORRs and DCRs between those two subgroups. Interestingly, combinations of anti–PD-1/PD-L1 and anti-VEGF therapy showed similar ORRs and DCRs in all HCCs, regardless of HBV status [177].

## 7. Predictive Biomarkers in HCC Immunotherapy

Identification of patients’ subgroups that would benefit from ICI remains a mainstay goal of cancer research and several biomarkers have been explored in solid tumors. Apart from PD-1/PD-L1 expression, another common predictor of tumor response is microsatellite instability (MSI). MSI refers to random mutations occurring in small repetitive elements due to a defective (mismatch repair) MMR system [178,179]. The accumulation of random mutations leads to increased neoantigen formation by the cells and target antigens for the immune system, promoting the expression of inflammatory cytokines and T-cell activation, thus rendering tumors susceptible to immunotherapy [180]. MSI high status (MSI-H) in HCC is rare, occurring in less than 3% [181,182,183,184,185]. Interestingly, another analysis, which included 122 patients with HCC, revealed no tumors displaying a typical MSI-H phenotype defined by PCR-based MSI testing [185]. Despite its rarity, inflammation-mediated dysfunction of the MMR pathway can contribute to the accumulation of mutations during hepatitis-associated tumorigenesis. Moreover, tumor mutational burden (TMB) analysis is useful as an agnostic histologic indicator to identify patients who can benefit from ICIs, and the use of PD-1 inhibitors is recommended in this setting as second or later line treatment [181]. In a study by Ang, 755 specimens of HCC were analyzed for biomarkers affecting response to PD-1/PD-L1 inhibitors. Out of 542 cases assessed for MSI, one (0.2%) was MSI-H and TMB high (TMB-H) [182]. Despite the low percentage of MSI-H, Kawaoka et al. reported encouraging results concerning response to immunotherapy. In their study, only two patients (2.4%) were detected MSI-H with advanced HCC, one of which had a complete response to pembrolizumab [186]. 

On the other hand, genetic profiling of HCC using next-generation sequencing (NGS) has provided new opportunities to extensively analyze and identify those patients with HCC likely to benefit from targeted therapies [187]. High-resolution copy-number analysis and whole-exome sequencing has led to the identification of key genes, such as *ARID1A*, *RPS6KA3*, *NFE2L2*, and *IRF2* and Wnt/β-catenin signaling pathway involvement in HCC tumors by oxidative stress metabolism and Ras/mitogen-activated protein kinase (MAPK) pathways [188]. Functional analyses showed tumor suppressor properties for *IRF2*, whose inactivation, exclusively found in HBV-related tumors, led to impaired *TP53* function [188]. β-catenin is shown to support HCC cell survival during the earlier stages of HCC by promoting EGFR signaling [189], while upregulation of LEF-1, a key transcription factor of β-catenin, is found also in hepatitis B surface antigen (HBsAg)-expressing HCC cell lines [190,191]. In addition, Wnt signaling supports more discrete functions, such as escaping immune surveillance. In the work of Harding et al., patients with advanced HCC were treated with anti-CTLA-4 monotherapy [*n* = 1], anti-PD-1/PD-L1 monotherapy [*n* = 25], and anti PD-1/PD-L1 plus other agents, including anti-CTLA-4 [*n* = 1], anti-LAG3 [*n* = 2], and anti-KIR [*n* = 2]). NGS analysis disclosed 10 patients with WNT/β-catenin mutations, none of whom had response to anti–PD-1 or anti–PD-L1 therapy at all, whereas 50% of *CTNNB1* WT patients had a response. WNT/β-catenin signaling pathway were associated with lower DCRs and lower median PFS (2.0 vs. 7.4 months; HR, 9.2; 95% CI, 2.9–28.8; *p* < 0.0001) and OS (9.1 vs. 15.2 months; HR, 2.6; 95% CI, 0.76–8.7; *p* = 0.11) compared with those without such alterations [18].

## 8. Conclusions

Despite advances in cancer treatment and translational research, HCC is still associated with inferior outcomes and high mortality. However, many clinical trials seek to evaluate the efficacy of immunotherapy in HCC, including ICIs, cancer vaccines, ACT, and combinations with chemoradiotherapy or other molecularly targeted agents, yielding some encouraging results. So far, immunotherapeutic strategies have been proved safe; however, studies of single agent ICIs failed to show a survival benefit compared to combination therapies. Importantly, combinations of immunotherapy with other modalities have resulted in better OS and PFS. The optimization of the best strategy remains challenging, mainly because of the low TMB and immune-suppressive environment on which HCC arises. Despite the encouraging results of a few MSI-H tumor cases that responded to immunotherapy, MSI-H phenotype seems to be a rare phenomenon in HCC. On the other hand, genetic profiling of HCC using NGS and identification of patients with HCC likely to benefit from immunotherapies has shown promising results. Further analysis of NGS data will soon allow a better understanding of tumor heterogeneity and its potential role in treatment decision making by identifying HCC patients likely to benefit from immunotherapies. This is also supported by recent results in WNT/β-catenin association with immune evasion and resistance to anti-PD-1 therapy. A low number of mutations hampers the production and release of neo-antigens and subsequently leads to low number of TILs. Additionally, HCC immunosuppressive TME creates a significant barrier to the efficacy of immunotherapy agents. Further research needs to focus on overcoming immunotherapy resistance by targeting multiple immune defects using combinatorial approaches of immunotherapy and cytotoxic agents in patients with HCC.

## Figures and Tables

**Figure 1 cancers-12-02859-f001:**
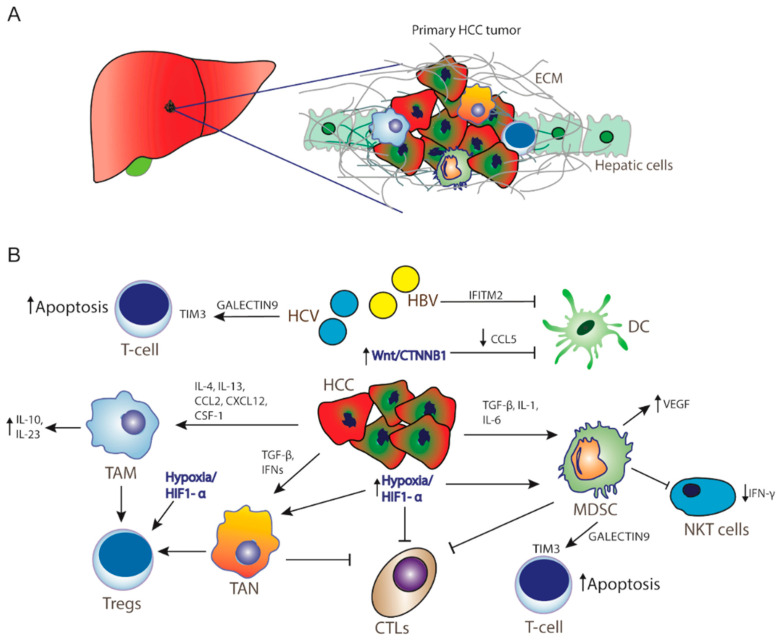
Hepatocellular carcinoma microenvironment. (**A**) Tumor microenvironment creates a barrier from extracellular matrix for immunotherapy as well as for cytotoxic drugs to act significantly in HCC. (**B**) Resistance mechanisms in HCC inducing tumor progression, immunosuppression, and cancer cell survival. Activation of Wnt/CTNNB1 signaling inhibits CCL5 production, thereby DC recruitment. Hypoxia and activation of HIF-1a promotes recruitment of MDSCs, Tregs, and TANs, whereas HCV infection and MDSCs promote T-cell apoptosis through activation of TIM3 receptor by Galectin-9 secretion by monocytes. HCC: hepatocellular carcinoma cells; ECM: extracellular matrix; TAM: tumor-associated macrophages; ΤAΝ: tumor-associated neutrophils; MDSCs: myeloid-derived suppressive cells; CTLs: cytotoxic T-lymphocytes; Tregs: CD4+ regulatory T cells; NKT cells: natural killer T cells; DC: dendritic cells IL: interleukin; TGF-β: transforming growth factor beta; IFNs: interferons; HiF1-a: hypoxia-inducible factor-1α; VEGF: vascular endothelial growth factor; CCL2: C-C motif chemokine ligand 2; CCL5: CC-chemokine ligand 5; CXCL12: C-X-C motif chemokine 12; CSF-1: colony stimulating factor 1; IFITM2: interferon-induced transmembrane protein 2.

**Table 1 cancers-12-02859-t001:** Immune checkpoint inhibitors—current clinical trials.

Intervention	Cancer Stage	Clinical Phase/Identifier	Progression Free Survival (PFS) (Months, 95% CI)	Median Overall Survival (OS) (Months, 95% CI)	Response Rates (%, 95% CI)	Bibliography
Nivolumab	Advanced HCC	Phase I/II, NCT01658878	3.4 (1.6–6.9), for DS 4.1 (3.7–5.5), for EX	15.0 (9.6–20.2), for DS NR, for EX	15% (6–28), for DS 20% (15–26), for EX	[52]
Nivolumab	Advanced HCC	Phase III, NCT02576509	3.7 (3.1–3.9)	16.4 (13.9–18.4)	15%	[53]
Sorafenib			3.8 (3.7–4.5)	14.7 (11.9–17.2) (HR 0.84, *p* = 0.0419)	7%	
Pembrolizumab, sorafenib	Advanced HCC	Phase II, NCT02702414	4.8 (3.4–6.6)	12,9 (9,7–15,5)	17% (11–26)	[54]
Pembrolizumab	Second-line, Advanced HCC	Phase III NCT02702401	3.0 (2.8–4.1)	13.9 (11.6–16.0)	18.3 (14.0–23.4)	[55]
placebo			2.8 (2.5–4.1)	10.6 (8.3–13.5) (HR 0.781, *p* = 0.023)	4.4 (1.6–9.4)	
Pembrolizumab, Lenvatinib	Unresectable HCC	Phase Ib	9.3 per mRECIST 8.6 per RECIST v1.1.	22.0	46.0% (36.0–56.3) per mRECIST 36.0% (26.6%–46.2) per RECIST v1.1	[56]
Atezolizumab, Bevacizumab	Unresectable HCC	Phase Ib NCT02715531	5.6 (3.6–7.4)			[57]
Atezolizumab			3.4 (1.9–5.2) (HR 0.55, *p* = 0.0108)			
Atezolizumab/Bevacizumab	Unresectable HCC	Phase III NCT03434379	6.8 (5.7–8.3)	67.2% (61.3–73.1)		[58]
Sorafenib			4.3 (4.0–5.6) (HR 0.59, *p* < 0.001)	54.6% (45.2–64.0) 12 months response		
Tremelimumab	HCC	Phase II NCT01008358	6.48 (3.95–9.14)		17.6%	[59]
Tremelimumab, RFA or TACE	Advanced HCC	Phase I/II NCT01853618	7.5 (5.6–9.3)	8.4 (6.5–10.3)		[60]

HCC: hepatocellular carcinoma; RFA: radiofrequency thermal ablation; TACE: transarterial chemoembolization; DS: dose-escalation group; EX: dose-expansion group; NR: not reached.

**Table 2 cancers-12-02859-t002:** Vaccine therapy—current clinical trials.

Intervention	Cancer Stage	Clinical Phase/Identifier	Progression Free Survival (PFS) (Months, 95% CI)	Median Overall Survival (OS) (Months, 95% CI)	Response Rates (%, 95% CI)	Bibliography	
GPC3-vaccine	Advanced HCC	Phase I, UMIN000001395	3.4 (2.1–4.6)	9.0 (8.0–10.0)	91%		[93]
GPC3-vaccine, Surgery and RFA	Adjuvant therapy	Phase II		20.1 (14.7–25.5)	1 year at 24%, 2 years at 52.4%		[94]
MRP3	HLA-A24-positive	Phase I UMIN000005678		14.0 (9.6–18.5)	72.7%		[95]
DCs	HCC patients	Phase II	6 months at 33%, 1 year at 11%	5.5			[96]
Ilixadencel	HCC patients	Phase I NCT01974661	5.5	7.4, for 1 0 × 10^6^ cells 11.8, for 20 × 10^6^ cells	73%		[97]
JX-594	Advanced HCC	Phase II NCT00554372		14.1, for high-dose 6.7, for low-dose	57%, for high-dose 67%, for low-dose		[98]
JX-594	Advanced HCC, previously treated with sorafenib	Phase IIb NCT01387555	1.8 (1.5–2.8)	4.2			[99]
BSC			2.8 (1.5–NA)	4.4 (HR, 1.19, *p* = 0.428)			

HCC: hepatocellular carcinoma; GPC3: glypican-3; MRP3: multidrug resistance-associated protein 3; DC: dendritic cell; BSC, best supportive care.

**Table 3 cancers-12-02859-t003:** Adoptive cell transfer—current clinical trials.

Intervention	Cancer Stage	Clinical Phase/Identifier	Progression free survival (PFS) (Months, 95% CI)	Median Overall Survival (OS) (Months, 95% CI)	Response Rates (%, 95% CI)	Bibliography	
CIK, TACE, and RFA	Advance HCC		17 (10.96–23.04)	56 (38.09–73.91)			[138]
TACE, RFA			10 (8.57–11.44)	31 (24.53–37.47)			
CAR-T cells, cyclophosphamide, and fludarabine	Advanced GPC3+ HCC (Child–Pugh A)	Phase I NCT02905188 NCT03146234	3.2 and 3.6 (for two patients)	9.1 (1.5–20)	Two partial responses		[151]
Adjuvant-adoptive immunotherapy	Adjuvant treatment, Resected HCC		48% (37–59)				[156]
control			33% (22–43)				

HCC: hepatocellular carcinoma; RFA: radiofrequency thermal ablation; TACE: transarterial chemoembolization; CIK: cytokine induced killer cells; CAR-T: chimeric antigen receptor T cells.

**Table 4 cancers-12-02859-t004:** Combination therapy—current clinical trials.

Intervention	Cancer Stage	Clinical Phase/Identifier	Progression Free Survival (PFS) (Months, 95% CI)	Median Overall Survival (OS) (Months, 95% CI)	Response Rates (%, 95% CI)	Bibliography
NIVO (1mg/kg), IPI (3mg/kg)	Sorafenib-treated advanced hepatocellular carcinoma patients	NCT01658878	54.0 (39.0–68.0)	23.0 (9.0–NA)	32%	[157]
NIVO (3mg/kg), IPI (1mg/kg)			43.0 (29.0–58.0)	12.0 (8.0–15.0)	31%	
NIVO (3mg/kg), IPI (1mg/kg)			49.0 (34.0–64.0)	13.0 (7.0–33.0)	31%	
NIVO, CABO	Sorafenib or experienced advanced hepatocellular carcinoma patients	NCT01658878	5.5	Not reached	81%	[158]
NIVO, IPI, CABO			6.8	Not reached	83%	
ATVAC	Resected, invasive HCC	UMIN000005820	24.5 (7.8–41.2)	97.7 (48.6–146.7)		[171]
Surgery alone			12.6 (6.9–18.3)	41.0 (16.3–65.8)		

NIVO: nivolumab; IPI: ipilimumab; CABO: cabozantinib, HCC: hepatocellular carcinoma; ATVAC: autologous tumor lysate-pulsed dendritic cell vaccine plus ex vivo activated T cell transfer.

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
