# Peer review of "Immunotherapy for Hepatocellular Carcinoma: A 2021 Update"

_cancers, 2020, doi:10.3390/cancers12102859_

Round 1

Reviewer 1 Report

The authors propose a review of the current status of immunotherapy treatment of hepatocellular carcinoma. In a rather peremptory way they title their review a 2021 update! They also propose in the abstract to explore the role of the microenvironment "as a predictive and prognostic marker to immunotherapy response and its clinical implications". I have to admit that these two points led me to accept the review, interested in an analysis of recent and innovative data in this field.

The review of known data up to the end of 2019 and the beginning of 2020 is fairly well conducted and exposed even if there is not much added value compared to other literature reviews that currently abound in this field.

Concretely, my main reservation about the publication of this article concerns important gaps in the data that are really current in 2020 or to come at the beginning of 2021. Similarly, the study of predictive factors and the microenvironment are only touched upon and important points are completely ignored.

Thus the non-exhaustive list of the most important points that I think are in this review and that must be addressed!

- No mention or analysis of the work of Daniela Sia et al (Gastroenterology 2017), Marina Ruiz de Galarreta et al (Cancer discovery 2019) and Ghassan Abou alfa et al (Clinical Cancer research 2019) regarding the impact of Wnt B catenin mutations on the lack of immune response and the potential impact on immunotherapy treatments.

- No mention or analysis of the work of Ghassan Abou alfaet al ( J Hepatol 2020) on the role of the microenvironment and the response to nivolumab in CM 040!

- Ongoing pivotal studies associating tyrosine kinase inhibitors and immunotherapies are not very well developed, even though these are the most expected results!!! (Lenvatinib + pembrolizumab, Cabozantinib and atezolizumab, etc...)

- In anti PD1 + CTLA4 combinations the latest results (ASCO 2020) are not discussed. The rationale and the different dose regimens as well as their clinical impact (tolerance and efficacy) are not discussed (ipilumumab, tremelemimab dosages and rhythm of administration).

For me it is important that in addition to the formal data fairly well known the authors also meet the expectation raised in the title and abstract.

Author Response

Reviewer #1:

Comments and Suggestions for Authors

The authors propose a review of the current status of immunotherapy treatment of hepatocellular carcinoma. In a rather peremptory way they title their review a 2021 update! They also propose in the abstract to explore the role of the microenvironment "as a predictive and prognostic marker to immunotherapy response and its clinical implications". I have to admit that these two points led me to accept the review, interested in an analysis of recent and innovative data in this field.

The review of known data up to the end of 2019 and the beginning of 2020 is fairly well conducted and exposed even if there is not much added value compared to other literature reviews that currently abound in this field.

Concretely, my main reservation about the publication of this article concerns important gaps in the data that are really current in 2020 or to come at the beginning of 2021. Similarly, the study of predictive factors and the microenvironment are only touched upon and important points are completely ignored.

Thus the non-exhaustive list of the most important points that I think are in this review and that must be addressed!

We thank the reviewer for his/her comments and helpful suggestions. Please see our point by point response below addressing reviewer’s concerns.

- No mention or analysis of the work of Daniela Sia et al (Gastroenterology 2017), Marina Ruiz de Galarreta et al (Cancer discovery 2019) and Ghassan Abou alfa et al (Clinical Cancer research 2019) regarding the impact of Wnt B catenin mutations on the lack of immune response and the potential impact on immunotherapy treatments.

AUTHOR RESPONSE AND ACTION TAKEN: We have included in the manuscript a paragraph (Lines 71-79) regarding the impact of Wnt/β-catenin pathway and the potential impact on immunotherapy treatments.

- No mention or analysis of the work of Ghassan Abou alfaet al ( J Hepatol 2020) on the role of the microenvironment and the response to nivolumab in CM 040!

AUTHOR RESPONSE AND ACTION TAKEN: We have included in the manuscript a paragraph (Lines 180-187) where we analyze the role of the microenvironment and the response to nivolumab in CM 040.

- Ongoing pivotal studies associating tyrosine kinase inhibitors and immunotherapies are not very well developed, even though these are the most expected results!!! (Lenvatinib + pembrolizumab, Cabozantinib and atezolizumab, etc...)

AUTHOR RESPONSE AND ACTION TAKEN: We have modified the manuscript including recent updates regarding tyrosine kinase inhibitors and immunotherapies. (Lines 227-237 and 257-263). Additional changes have been incorporated in table 1.

- In anti PD1 + CTLA4 combinations the latest results (ASCO 2020) are not discussed. The rationale and the different dose regimens as well as their clinical impact (tolerance and efficacy) are not discussed (ipilumumab, tremelemimab dosages and rhythm of administration).

AUTHOR RESPONSE AND ACTION TAKEN: We have included in the section “Combinations strategies of immunotherapies”, a paragraph (Lines 452-472) where we analyze the recent results (ASCO 2020) regarding the rationale and the different dose regimens as well as their clinical impact. We have also updated table 4 accordingly

For me it is important that in addition to the formal data fairly well known the authors also meet the expectation raised in the title and abstract.

Submission Date

14 August 2020

Date of this review

25 Aug 2020 22:52:29

Reviewer 2 Report

Summary:

This review focuses on immuntherapy in hepatic cell carcinoma (HCC). The review begins with a brief introduction to immunotherapy and to HCC, detailing the immunosuppressive environment of HCC and the modalities of immunotherapy. Next, current immune checkpoint inhibitors and clinical trials that are ongoing in HCC are discussed, followed by cancer vaccine trials. Finally, adoptive cell transfer regimens in HCC immunotherapy and combinations of immunotherapies that are discussed. In addition to the modalities of immunotherapy, the authors address a very important point when treating patients with HCC – inflammation due to having the co-morbidities of active hepatitis infections in many HCC cases. The authors then identify another strong interest in immunotherapy research right now, neoantigens, and discuss that MSI high and TMB- high patients are very few in HCC cases. Lastly, they identify how the field is attempting to increase immunotherapy efficacy for HCC.

Strengths:

The authors did an outstanding job at reviewing the ongoing and completed clinical trials for HCC immunotherapies. The tables are clear and useful. Overall, the review is well written and organized.

Weaknesses:

Figure 1 is appreciated, but there are several things in the figure that are not mentioned in the text, including hypoxia and other cytokines. It may be useful to show how hepatitis infection influences these immune cells in the figure.

One piece that is missing is how immunotherapies are being combined with standard therapies such as sorafinib.

Also, the authors could comment on whether tumor genotyping, NGS, TMB etc are being used clinically or preclinically to try to define HCC tumors that will respond to immunotherapies.

Minor points (grammar etc):

The rationale for treating patients with sorafinib or other TKIs is not described.

For Nivolumab, the Child-Pugh score is mentioned but it is not clear what that is and why it is used.

It is mentioned that liver endothelium expresses PD-L1, but do the liver carcinoma cells themselves express PD-L1? Which epithelial cells do HCCs originate from?

Line 413, the 1 is missing from PD1.

Line 421, the anti is missing from anti-PD-L1.

The opening sentence - HCC is the most common liver cancer AND is associated with poor prognosis, or is the most common poor prognosis liver cancer?

Introduction, by 2030, more than 1 million patients will die - between now, 2020, and 2030? Or all time? Perhaps rewrite to "in the next 10 years"?

Author Response

Reviewer #2:

Comments and Suggestions for Authors

This review focuses on immuntherapy in hepatic cell carcinoma (HCC). The review begins with a brief introduction to immunotherapy and to HCC, detailing the immunosuppressive environment of HCC and the modalities of immunotherapy. Next, current immune checkpoint inhibitors and clinical trials that are ongoing in HCC are discussed, followed by cancer vaccine trials. Finally, adoptive cell transfer regimens in HCC immunotherapy and combinations of immunotherapies that are discussed. In addition to the modalities of immunotherapy, the authors address a very important point when treating patients with HCC – inflammation due to having the co-morbidities of active hepatitis infections in many HCC cases. The authors then identify another strong interest in immunotherapy research right now, neoantigens, and discuss that MSI high and TMB- high patients are very few in HCC cases. Lastly, they identify how the field is attempting to increase immunotherapy efficacy for HCC.

Strengths:

The authors did an outstanding job at reviewing the ongoing and completed clinical trials for HCC immunotherapies. The tables are clear and useful. Overall, the review is well written and organized.

We thank the reviewer for his/her comments and helpful suggestions. Please see our point by point response below addressing reviewer’s concerns.

Weaknesses:

- Figure 1 is appreciated, but there are several things in the figure that are not mentioned in the text, including hypoxia and other cytokines. It may be useful to show how hepatitis infection influences these immune cells in the figure.

AUTHOR RESPONSE AND ACTION TAKEN: We have modified the text of the figure 1 accordingly (lines 93-106), including abbreviations that were mentioned in the figure but not in the text. Moreover, an updated figure is provided including the effect of hepatitis infection on immune cells.

- One piece that is missing is how immunotherapies are being combined with standard therapies such as sorafinib.

AUTHOR RESPONSE AND ACTION TAKEN: We have included and analyzed the results of clinical trials regarding treatment with TKIs, such as lenvatinib and cabozantinib alone or in combination with immunotherapy, while additional references are included in the tables provided (Lines 227-237 and 257-263).

- Also, the authors could comment on whether tumor genotyping, NGS, TMB etc are being used clinically or preclinically to try to define HCC tumors that will respond to immunotherapies.

AUTHOR RESPONSE AND ACTION TAKEN: We have included a paragraph (Lines 575-593) as well in the discussion (Lines 604-610) where we analyze the use of genotyping and NGS and latest findings in HCC tumor response to immunotherapy.

Minor points (grammar etc):

- The rationale for treating patients with sorafinib or other TKIs is not described.

AUTHOR RESPONSE AND ACTION TAKEN: A short paragraph in the introduction section was added where we address the rationale for treating patients with TKIs (Lines 115-120).

- For Nivolumab, the Child-Pugh score is mentioned but it is not clear what that is and why it is used.

AUTHOR RESPONSE AND ACTION TAKEN: We have included a short paragraph where we address this issue (Lines 172-174)

- It is mentioned that liver endothelium expresses PD-L1, but do the liver carcinoma cells themselves express PD-L1? Which epithelial cells do HCCs originate from?

AUTHOR RESPONSE AND ACTION TAKEN: We have included a short paragraph where we address this issue (Lines 71-74). Hepatocellular carcinomas originate predominantly from hepatocytes and non-cancerous lesions (regenerative nodules and adenomas) from hepatic progenitor cells. Increased PD-1 and PD-L1 expression has been observed in HCC patients, with the expression of PD-L1 associated with tumor aggressiveness and poor prognosis

- Line 413, the 1 is missing from PD1.

AUTHOR RESPONSE AND ACTION TAKEN: We have corrected the typing error (Line 515)

- Line 421, the anti is missing from anti-PD-L1.

AUTHOR RESPONSE AND ACTION TAKEN: We have corrected the typing error (Line 523)

- The opening sentence - HCC is the most common liver cancer AND is associated with poor prognosis, or is the most common poor prognosis liver cancer?

AUTHOR RESPONSE AND ACTION TAKEN: We have revalidated the published bibliography and we confirm that HCC is referred as the most common liver malignancy associated with poor prognosis.

- Introduction, by 2030, more than 1 million patients will die - between now, 2020, and 2030? Or all time? Perhaps rewrite to "in the next 10 years"?

AUTHOR RESPONSE AND ACTION TAKEN: We have rewritten the text according to the suggestions (Lines 43).

Submission Date

14 August 2020

Date of this review

15 Sep 2020 15:44:16

Reviewer 3 Report

Authors provided an updated overview of immunotherapy in HCC. A detailed information on current status of various immunotherapy options was nicely presented. I have some minor suggestions as detailed below.

  1. Figure 1 – please label the cell types in the image, alternatively provide a key at the bottom of the image.
  2. Table 2, vaccine therapy – correct the typo in patients.

Author Response

Reviewer #3:

We thank the reviewer for his/her comments and helpful suggestions. Please see our point by point response below addressing reviewer’s concerns.

Comments and Suggestions for Authors

Authors provided an updated overview of immunotherapy in HCC. A detailed information on current status of various immunotherapy options was nicely presented. I have some minor suggestions as detailed below.

  1. Figure 1 – please label the cell types in the image, alternatively provide a key at the bottom of the image.

AUTHOR RESPONSE AND ACTION TAKEN: We have modified the Figure 1 and the text of the figure 1 accordingly (lines 93-106), including abbreviations that were mentioned in the figure but not in the text.

  1. Table 2, vaccine therapy – correct the typo in patients.

AUTHOR RESPONSE AND ACTION TAKEN: We have performed the correction in the text in table 2 (lines 301-302)

Submission Date

14 August 2020

Date of this review

10 Sep 2020 21:38:11

Round 2

Reviewer 1 Report

I'm agree with all the modifications